# Virus-Specific Defense Responses in Sweetpotato: Transcriptomic Insights into Resistance and Susceptibility to SPFMV, SPCSV, and SPVD

**DOI:** 10.3390/biology14111541

**Published:** 2025-11-03

**Authors:** Joanne Adero, Reuben Ssali, Fuentes Segundo, David Maria, Mercy Kitavi, Benard Yada, Denis Karuhize Byarugaba, Faruk Dube, Peace Proscovia Aber, Stephen Obol Opiyo, Zhangjun Fei, Jan Frederik Kreuze

**Affiliations:** 1National Crops Resources Research Institute, National Agricultural Research Organization, Kampala P.O. Box 7084, Uganda; 2College of Veterinary Medicine, Animal Resources and Biosecurity, Makerere University, Kampala P.O. Box 7062, Uganda; 3International Potato Center (CIP), Plot 47, Ntinda II Road, Kampala P.O. Box 22274, Uganda; 4Virology Laboratory, Crop and Systems Science Division, International Potato Center (CIP), Lima 12, Peru; 5Research Technology Support Facility, Michigan State University, East Lansing, MI 48824, USA; 6Department of Medical Biochemistry and Microbiology, Uppsala University, BMC, Husargatan 3, 752 37 Uppsala, Sweden; 7TeraSiq Analytics Limited, Kampala P.O. Box 184125, Uganda; 8MAGMA Consultants International, Kampala P.O. Box 75180, Uganda; 9Faculty of Management Science, University of Sacred Heart, Gulu P.O. Box 374, Uganda; 10Patira Data Science, Kampala P.O. Box 12464, Uganda; 11Boyce Thompson Institute, Cornell University, Ithaca, NY 14853, USA; 12International Potato Centre, CIP Headquarters Lima, Avenida La Molina 1895, La Molina Apartada 1558, Lima 15024, Peru; j.kreuze@cgiar.org

**Keywords:** sweetpotato, sweet potato chlorotic stunt virus (SPCSV), sweet potato feathery mottle virus (SPFMV), sweet potato virus disease (SPVD), transcriptome profiling

## Abstract

Sweetpotato is a key food crop worldwide, but its production is under threat from viral diseases, especially one called sweet potato virus disease which occurs when two different viruses infect the plant at the same time. This study examined how three types of sweetpotato plants—‘Beauregard’, ‘Tanzania’, and ‘New Kawogo’—respond to infection with each virus on its own and to the combined disease. By studying plant responses at different times after infection, the researchers discovered that ‘New Kawogo’ was able to trigger strong and lasting defenses that helped it tolerate infection, while ‘Beauregard’ showed weak and delayed defenses, making it more vulnerable. The research identified specific genes linked to either resistance or susceptibility, which can be used to guide the development of new sweetpotato varieties that are better able to withstand viral diseases. These findings are important because they provide a clear understanding of how different sweetpotato varieties naturally fight off viruses, and they open the door to breeding stronger, more resilient crops. This can help to secure sweetpotato harvests, improve food security, and support farmers who rely on this crop for their livelihoods.

## 1. Introduction

Sweetpotato (*Ipomoea batatas* L. Lam) is a crucial global food crop, particularly in sub-Saharan Africa where it significantly contributes to food, income, and nutritional security. Particularly, orange-fleshed sweetpotato (OFSP) varieties are widely grown by smallholder women farmers. OFSPs have a high β-carotene content—a precursor for vitamin A which is essential for immune function, vision, skin health, and most importantly in combating vitamin A deficiency in sub-Saharan Africa (SSA) [1]. This deficiency affects nearly 40% of children under five in the region. In addition to OFSP, purple-fleshed varieties offer health benefits such as anti-inflammatory effects, reduced oxidative stress, and lower risk of chronic diseases like heart disease and cancer because they are rich in anthocyanins and phenolic compounds [2,3]. Despite its global production reaching nearly 90 million metric tons [4], viral diseases remain a major limitation, with sweet potato virus disease (SPVD) as the most economically important disease causing yield losses of up to 98% and accounting for cultivar degeneration [5]. SPVD is characterized by severe plant stunting, the development of small, narrow, distorted, or crinkled leaves, chlorotic (yellow) mottling or mosaic patterns on the leaves, and vein clearing. In single infections, SPFMV may present mild or no symptoms in sweetpotato plants. During co-infection with *Sweet potato chlorotic stunt virus* (SPCSV; family *Closteroviridae,* genus *Crinivirus*), *Sweet potato feathery mottle virus* (SPFMV) symptoms become severe and result in sweet potato virus disease (SPVD) syndrome.

This arises from the synergistic interaction between SPCSV and SPFMV [6]. SPVD is spread worldwide and in Uganda, prevalence exceeds 90% in some regions, highlighting the urgent need for resistant cultivars [7]. However, conventional breeding efforts are hindered by the crop’s hexaploid genome, clonal propagation, and both self- and cross-incompatibility factors that significantly limit the speed of genetic improvement [8,9]. These challenges underscore the need for genomics-based breeding strategies, which offer powerful and efficient alternatives for sweetpotato improvement. Unfortunately, the lack of critical genomic and genetic resources such as a fully annotated reference genome continues to limit our understanding of host–virus interactions [8,9] and restricts the application of marker-assisted selection for traits like SPVD resistance. Developing these resources is essential to uncover the molecular basis of trait variation and to accelerate the design and implementation of effective marker-assisted breeding strategies [10,11].

Recently, high-quality de novo whole-genome assemblies for two diploid relatives of sweetpotato, *Ipomoea trifida* and *Ipomoea triloba* (wild ancestor of sweetpotato), have been reported and have shown to be robust references for sweetpotato which is a hexaploid relative [12]. These assemblies are expected to support studies of sweetpotato and enhance the development of marker-assisted breeding programs by identifying regions or genes that are involved in important traits such as disease resistance, root yield, and nutrient content [13]. *I. trifida* transcriptomics has been used in several studies [14] and will be used in our study to identify genes of disease resistance in hexaploid sweetpotato. Recent studies have increasingly applied transcriptomic approaches to investigate sweetpotato responses to viral infections, particularly SPVD. For instance, Bednarek et al., 2021 [15] identified key defense-related genes and disruptions in hormone signaling under SPVD stress. However, these studies were limited to single timepoints or specific genotypes, which restricts their broader applicability. Earlier transcriptomic by McGregor et al., 2009 [16] provided foundational insights into sweetpotato virus interactions but lacked the resolution and sensitivity of next-generation sequencing (NGS) technologies.

RNA sequencing (RNA-seq), an NGS-based approach, has transformed the study of host–pathogen interactions by enabling the comprehensive identification of differentially expressed genes (DEGs), even in genetically complex and under-resourced crops like sweet potato [17]. It offers a high-resolution view of transcriptomic changes, allowing for more precise insights into host immune responses.

Building on previous work [15], this study employs time-series RNA-seq profiling across three phenotypically distinct sweet potato cultivars in response to SPVD-causing viruses. The goal was to generate a comprehensive transcriptomic profile and identify differentially expressed genes (DEGs) associated with SPCSV, SPFMV, and SPVD infection. This analysis will enhance our understanding of the molecular mechanisms underlying sweetpotato responses to SPVD viruses and other virus-induced biotic stresses, while also uncovering potential candidate genes involved in resistance or susceptibility. Unlike traditional microarray approaches, the use of RNA-seq in this study provides higher resolution and broader coverage, enabling a more detailed and accurate assessment of gene expression dynamics during viral infection. This study focused on three sweetpotato cultivars: ‘Beauregard’ (a susceptible cultivar), ‘Tanzania’ (moderately tolerant), and ‘New Kawogo’ (highly tolerant). This design enables the capture of both temporal and genotype-specific gene expression dynamics in response to SPFMV, SPCSV, and SPVD infections, analyzed independently. Furthermore, the study identifies candidate genes linked to resistance and susceptibility and proposes molecular markers for potential integration into marker-assisted breeding strategies. Collectively, this work represents a significant advancement in sweetpotato functional genomics and contributes valuable insights into virus-specific immune responses.

## 2. Materials and Methods

### 2.1. Infection Experiments

The infection time-series experiment was conducted under controlled conditions in a greenhouse. *Ipomoea setosa,* a wild relative highly susceptible to sweetpotato viruses, was planted and was first graft-inoculated with the viruses to serve as an infected source plant with a high viral load using (i) *Sweet potato feathery mottle virus* (SPFMV; Piu3 isolate, GenBank ID: FJ155666), (ii) *Sweet potato chlorotic stunt virus* (SPCSV; m2_47 isolate, GenBank IDs: HQ291259 for RNA1 and HQ291260 for RNA2), or (iii) both viruses combined to simulate SPVD infection.

Scions from these virus-infected *I. setosa* plants were then grafted onto healthy plants from three cultivars: ‘Beauregard’, ‘New Kawogo’, and ‘Tanzania’.

As a control (mock treatments), healthy plants from each cultivar were grafted with virus-free *I. setosa* scions. All plants were maintained in greenhouse conditions throughout the study. Leaf samples were collected from the 3rd, 4th, and 5th fully expanded leaves (counted from the apex) at 3-, 6-, and 12-weeks post-inoculation (WPI). Each treatment group included three biological replicates per cultivar, resulting in a total of 108 samples (3 virus treatments and mock treatment) × 3 timepoints × 3 replicates × 3 cultivars). Total RNA was extracted from leaf samples using the TRIzol (Invitrogen, Carlsbad, CA, USA) method [18]. RNA-seq libraries were prepared according to the protocol described in [19].

### 2.2. Viral Transcripts and vsiRNAs in Infected Plants

This study investigated the transcriptional dynamics of sweetpotato in response to infections with SPCSV, SPFMV, and their co-infection, SPVD, across the three cultivars. The small RNA (sRNA) profiles associated with sweetpotato viruses were processed. Raw sRNA sequencing used a streamlined protocol for sRNA library preparation and sequencing, drawing on methodologies similar to those described by Bednarek et al., 2021 [15]. After total RNA extraction from sweetpotato leaf samples, the sRNA fraction (18–30 nucleotides) was enriched through size-selection gel electrophoresis, targeting sequences critical for viral RNA silencing. Adapters and trailing bases were trimmed from raw reads using a script from the VirusDetect package [20]. The cleaned reads were aligned to the SILVA rRNA sequence databases [21] using Bowtie 2.3.3 [22] with up to one mismatch allowed, and reads mapping to rRNA were discarded.

To study viral sRNAs, reads were mapped to the SPCSV isolate m2-47 and SPFMV isolate Piu3 using Geneious Prime version 2019.2, with an allowance for one mismatch to ensure accurate mapping of virus-derived sRNAs, mirroring the alignment strategy employed for Figure 2c,d as in Bednarek et al., 2021 [15].

### 2.3. RNA-Seq Data Processing and Transcriptome Sequencing

RNA-seq libraries were constructed following the methodology outlined in Zhong et al., 2011 [19]. Briefly, mRNA was enriched by magnetic beads with Oligo (dT) and then sheered into short fragments in the fragmentation buffer. First-strand cDNA was synthesized using SuperScriptIII reverse transcriptase (Invitrogen) in the presence of dNTPs. Second-strand cDNA was synthesized using RNase H (NEB) and the Klenow fragment of DNA polymerase I (NEB) with a dUTP mix at 16 °C for 2.5 h using dUTPs. After end repair, adapters were ligated to the double-stranded cDNA, and the dUTP-containing strand was degraded using a uracil DNA glycosylase. RNA-seq libraries were sequenced on an Illumina HiSeq 2500 system manufactured by Illumina, Inc., based in San Diego, CA, USA. Raw RNA-seq reads for ‘Beauregard’ were deposited in the NCBI Sequence Read Archive (SRA) under the accession number PRJNA649319. Raw RNA-seq reads for ‘New Kawogo’ and ‘Tanzania’ were uploaded onto a server at Cornell University under the RNA-seq data processing for the Genomic Tools for Sweet Potato Improvement (GT4SP) project, funded by the Bill & Melinda Gates Foundation (Contract ID: OPP1052983), for analysis.

RNA sequencing data processing followed a standardized computational pipeline using nf-core/rnaseq v3.18.0 [23] executed with Nextflow v24.10.4. Raw sequencing reads underwent quality assessment using FastQC v0.12.1 [24] followed by quality trimming and adapter removal using fastp v0.23.4 [25]. Ribosomal RNA contamination was removed using SortMeRNA v4.3.7 [26] against the SILVA SSU, LSU v111 [21] and RFAM 5/5.8S v11.0 [27] databases. Expression quantification was performed against the diploid *I. trifida* transcriptome using the NSP306 v3 reference genome assembly [12]. Use of a single common reference genome enabled unbiased cultivar comparisons by ensuring all three sweetpotato cultivars were mapped to identical genomic coordinates, eliminating potential reference-induced artifacts in cross-cultivar differential expression analysis. Transcript-level abundance estimation was conducted using Salmon v1.10.3 [28] in quasi-mapping mode. Complementary read alignment was performed using STAR v2.7.11b [29] in two-pass mode, with alignment statistics computed using Samtools v1.21 [30]. Quality control metrics were evaluated using RSeQC v5.0.2 [31], Qualimap v2.3 [32], and dupRadar v1.32.0. Salmon transcript abundance estimates were imported using tximeta v1.20.1 [33] and summarized using SummarizedExperiment v1.32.0 in R version 4.0.3. Genes with fewer than 20 counts in a minimum of 3 samples were filtered to ensure robust statistical inference across the complete factorial experimental design encompassing 3 cultivars (‘Beauregard’, ‘Tanzania’, ‘New Kawogo’), 4 treatments (Mock, SPCSV, SPFMV, SPVD), and 3 timepoints (3, 6, 12 WPI).

### 2.4. Statistical Analysis

Count data normalization and statistical modeling were performed using DESeq2 [34] with a full factorial model incorporating main effects and two-way interactions:~*Cultivar* + *Treatment* + *Timepoint* + *Cultivar*/*Treatment* + *Cultivar*:*Timepoint* + *Treatment*/*Timepoint*

This design enabled assessment of main effects, cultivar-specific responses to viral infection, temporal variation in cultivar responses, and time-dependent viral effects. Variance-stabilized transformation was applied to account for experimental design during transformation. Principal component analysis was conducted using the 500 most variable genes to assess sample clustering and identify potential batch effects.

Differential expression analysis addressed two distinct biological hypotheses through systematic factor releveling approaches. Viral effects within cultivar–time combinations were assessed through direct comparisons between each viral treatment and mock controls, generating 27 pairwise comparisons (3 cultivars × 3 timepoints × 3 viruses). Cultivar-specific viral response differences were identified through interaction effects analysis comparing viral effects between cultivars within each timepoint using a difference-in-differences approach.

For each baseline cultivar–timepoint combination, factor levels were systematically releveled and DESeq2 modeling repeated to extract specific contrast coefficients. The negative binomial generalized linear model was fitted using the Wald test with 5000 maximum iterations (nbinomWaldTest, maxit = 5000) to ensure convergence for complex interaction terms. Log_2_ fold change estimates were improved using adaptive shrinkage (lfcShrink with type = ‘ashr’) to provide accurate effect size estimates. Multiple testing correction was applied using the Benjamini–Hochberg method at α = 0.05. Differentially expressed genes were defined as those with an adjusted *p*-value < 0.05 and absolute log_2_ fold change ≥ 0.5, following established recommendations for RNA-seq analysis in plant systems [35].

### 2.5. Functional Annotation and Enrichment Analysis

Orthologous gene relationships were established between sweetpotato (NSP306_trifida_v3) and *Arabidopsis thaliana* (Athaliana_167) using OrthoFinder v2.5.5 [36] to not only overcome limited functional annotation resources but also allow cross-cultivar comparison. Primary transcript protein sequences were extracted from both species, with *Arabidopsis* proteins obtained from TAIR10/Araport11 annotation version 167 operated by Phoenix Bioinformatics, located in Pleasanton, CA, USA and hosted by the Arabidopsis Information Portal [Araport], managed by the J. Craig Venter Institute (JCVI), located in Rockville, MD, USA. OrthoFinder analysis [36] used default parameters to identify orthogroups based on sequence similarity and phylogenetic relationships. Many-to-many relationships within orthogroups were processed by treating each mapping as a separate entry to maximize functional annotation coverage while preserving original differential expression statistics. Gene ontology overrepresentation analysis (ORA) was performed using the gprofiler2 R package version 0.1.0 [37], with DEGs mapped to corresponding *Arabidopsis* orthologs. ORA was conducted separately for upregulated and downregulated gene sets within each experimental condition (cultivar × virus × timepoint combination) across the biological process, molecular function, and cellular component GO domains. A custom background gene set was constructed from all sweet potato genes with valid *Arabidopsis* ortholog mappings that passed expression filtering. Statistical significance was assessed using g:SCS multiple testing correction at *p* < 0.05, with terms flagged as ‘driver terms’ prioritized for interpretation as the most statistically robust and biologically relevant enrichments.

Defense-related pathways were defined using literature-curated GO terms critical in plant antiviral defense mechanisms. Eleven evidence-based pathways encompassed primary antiviral mechanisms including RNA silencing and miRNA-mediated regulation [38], hormone signaling pathways including salicylic acid and jasmonic acid/ethylene responses [39], resistance mechanisms including NBS-LRR resistance [32,34,36] and pattern recognition, cellular defense processes including RNA decay, oxidative stress response, and cell wall defense, and metabolic responses including secondary metabolism and protein quality control [39]. Defense genes were extracted using dual approaches combining direct GO term matching with GO name pattern matching through regular expressions, restricted to significantly differentially expressed genes (padj < 0.05, |log_2_FC| ≥ 0.5). Pathway-level expression profiles were calculated by averaging variance-stabilized expression values of constituent genes, with analysis restricted to pathways containing a minimum of 5 genes to ensure robust pathway-level estimates.

### 2.6. Identification of Potential Molecular Markers Underlying Viral Resistance and Susceptibility

To identify candidate molecular markers associated with viral resistance and susceptibility in the sweetpotato cultivars ‘Beauregard’ and ‘New Kawogo’, differentially expressed genes (DEGs) were analyzed across 3, 6, and 12 WPI. Genes consistently upregulated or downregulated across timepoints were prioritized for further evaluation. Functional annotation linked these DEGs to biological processes indicative of host defense or vulnerability. The moderately tolerant cultivar ‘Tanzania’ was excluded from this analysis to allow for a clearer comparison between distinctly resistant (‘New Kawogo’) and susceptible (‘Beauregard’) responses. This binary contrast enabled a more precise identification of molecular signatures that distinctly separate effective resistance mechanisms from vulnerability.

## 3. Results

### 3.1. Symptoms of Sweetpotato Plants Infected

Co-infection of SPFMV and SPCSV showed the highest symptoms in all cultivars. Symptoms observed in plants included vein clearing, mosaic, roll down, leaf deformation, and chlorotic stunting (Figure 1A). ‘Beauregard’ showed the highest number of plants with symptoms (vein clearing, leaf deformation, roll down and mosaic) at all timepoints with SPVD infection. Infections with SPFMV and SPSCV exhibited no symptoms at 3 WPI; however, plants developed leaf roll down at 6 WPI. At 12 WPI, partial recovery was observed for SPCSV, but symptoms remained unclear for SPFMV infection (Figure 1B). ‘Tanzania’ exhibited moderate symptoms (vein clearing) at 3 WPI in all infections. Complete recovery was observed for SPFMV infection at 6 to 12 WPI. Chlorotic stunting was exhibited in plants inoculated with SPCSV and SPVD at 6 WPI, and there was full recovery at 12 WPI (Figure 1C). ‘New Kawogo’ had the least number of symptoms in two plants at 3 and 6 WPI only in SPVD infection and full recovery at 12 WPI, with no symptoms observed in SPFMV and SPCSV infection (Figure 1D).

### 3.2. Small RNA Mapping to Viral Genomes in Sweetpotato

To investigate responses to viral infections in sweetpotato, we analyzed the percentage of sRNA reads mapped to the genomes of SPCSV and SPFMV across two cultivars (‘New Kawogo’ and ‘Tanzania’), four treatments (mock, SPFMV, SPCSV, and SPVD), and three timepoints (3, 6, and 12 WPI). sRNA detection for ‘Beauregard’ are published by Bednarek et al., 2021 [15]. Bar charts were constructed to visualize the mean percentage of sRNA reads mapped to SPCSV and SPFMV genomes for each condition (Figure 2). In the ‘New Kawogo’ cultivar, SPVD-infected plants showed the highest sRNA mapping at 6 WPI to both SPFMV (3.04% mapped reads) and SPCSV (3.99%). In SPCSV-infected plants, the cultivar retained elevated levels of sRNA across all timepoints (2.17–2.48%) (Figure 2A, left panel). SPFMV-infected plants exhibited moderate SPFMV mapping (0.98–1.85%), with minimal SPCSV mapping (<0.17%). In the ‘Tanzania’ cultivar, SPCSV mapping was highest in SPCSV-infected plants at 6 WPI (5.22%), followed by SPVD at 6 WPI (4.31%) (Figure 2B, right panel). SPFMV mapping peaked in SPFMV-infected plants at 6 WPI (2.33%) and in SPVD at 12 WPI (1.89%). These results indicate that SPCSV induces stronger sRNA responses than SPFMV, particularly at 6 WPI, with ‘Tanzania’ showing more pronounced viral sRNA accumulation than ‘New Kawogo’. The time- and cultivar-specific patterns underscore the dynamic role of sRNAs in sweetpotato’s antiviral defense mechanisms. Mock treatments had near-zero viral mapping, as expected showing negligible mapping to both viruses (<0.05%) likely a result of sample bleeding [40] from virus infected samples multiplexed in the same sequencing lane as has been reported for Illumina sequencing. Regarding ‘Tanzania’, mock-treated ‘Tanzania’ plants displayed low mapping to both viruses (<0.32%). ‘New Kawogo’ had moderate SPFMV mapping (0.98–1.85%), suggesting weaker silencing.

### 3.3. Gene Expression Following Infection

Transcriptomics reveal distinct cultivar and treatment clustering patterns. To characterize viral infection responses, we sequenced total RNA from three cultivars (‘Beauregard’, ‘New Kawogo’, and ‘Tanzania’) at 3, 6, and 12 WPI with SPFMV, SPCSV, and SPVD. Sequencing generated an average of 12.5 million read-pairs per sample. Quality control using fastp and SortMeRNA produced high-quality reads with mean Phred scores above 33. STAR alignment to the diploid *I. trifida* NSP306 v3 reference transcriptome achieved 82% average mapping rates [12].

#### 3.3.1. Principal Component Analysis (PCA)

PCA explained 44% of the total variance (PC1: 23%, PC2: 21%) and revealed distinct clustering patterns driven by cultivar, timepoint, and treatment as shown in Figure 3. ‘Beauregard’ exhibited the most divergent transcriptional profiles with the greatest sample dispersion. ‘Tanzania’ and ‘New Kawogo’ formed tighter clusters, with cultivar-specific differences most pronounced at 12 weeks (Figure 3A). Treatment clustering showed temporal progression along PC1 and treatment separation along PC2. Mock controls formed distinct clusters at all timepoints (Figure 3B). At 3 weeks, viral treatments clustered in the upper right quadrant. By 6 weeks, all treatments shifted to negative PC2 values. At 12 weeks, viral treatments showed increased PC1 dispersion while maintaining separation from controls. SPVD samples were positioned intermediately between single virus infections.

#### 3.3.2. Hierarchical Clustering Identifies Discrete Transcriptional Response Modules

To identify coordinated gene expression patterns, we performed unsupervised hierarchical clustering of the 50 most statistically significant DEGs (ranked by adjusted *p*-value). Analysis revealed discrete transcriptional response modules with distinct expression architectures across treatment comparisons (Figure 4). Row-scaled log_2_ fold change matrices showed bifurcation into two primary gene clusters: an upper module with consistent transcriptional activation across viral treatments (positive log_2_FC values) and a lower module with complex expression dynamics including both induction and repression events. Column-wise clustering stratified comparisons according to treatment modality and temporal kinetics. Viral infection conditions manifested coordinated differential expression signatures distinct from mock-inoculated controls. SPVD co-infection conditions exhibited intermediate transcriptional profiles. Cultivar-dependent response heterogeneity was prominent, with discrete gene modules displaying enhanced sensitivity to specific genotypic backgrounds across the infection time course.

#### 3.3.3. UpSet Analysis Reveals Complex DEG Sharing Patterns Across Experimental Conditions

Comprehensive UpSet intersection analysis across 27 condition-specific DEG sets characterized overlap architectures reflecting both conserved and specialized transcriptional responses (Figure 5). SPFMV_‘Tanzania’_6 (780 DEGs) and SPVD_‘New Kawogo’_12 (366 DEGs) were the most transcriptionally responsive conditions. Condition-wise DEG enumeration displayed a substantial dynamic range, spanning from >700 genes in highly responsive conditions to <20 genes in minimally perturbed states.

Intersection topology analysis revealed both core viral response signatures and condition-specific transcriptional modules. Predominant intersection patterns involved 2–4 conditions. The largest multi-condition intersection comprised genes consistently dysregulated across diverse viral treatments and sampling timepoints. Several conditions exhibited minimal intersection overlap. The ‘Tanzania’ cultivar dominated the highest-magnitude gene sets, while smaller intersections distributed across cultivars and treatments, revealing hierarchical organization of shared versus unique transcriptional responses.

#### 3.3.4. Temporal Analysis Demonstrates Virus-Cultivar-Specific Responses

Longitudinal analysis of DEG abundance revealed pronounced virus–cultivar interaction effects across the infection timeline (Figure 6). ‘Tanzania’ exhibited the most robust and sustained transcriptional responses. SPCSV and SPVD maintained elevated DEG counts (log_10_ ≈ 3.0–3.5) throughout the experimental time course for both transcriptionally activated and repressed gene sets. SPFMV in ‘Tanzania’ displayed declining kinetic profiles, transitioning from moderate initial activation to near-baseline levels by 12 weeks. ‘Beauregard’ demonstrated distinctive biphasic response kinetics with initial transcriptional contraction at 6 weeks followed by secondary activation. Both SPCSV and SPVD infections exhibited analogous U-shaped temporal trajectories, declining from ~2.5–2.8 log_10_ DEGs at 3 weeks to ~1.0–1.5 at 6 weeks, subsequently recovering to ~2.5 by 12 weeks. SPFMV in ‘Beauregard’ showed delayed response kinetics, remaining transcriptionally quiescent until 12 weeks when substantial activation occurred.

‘New Kawogo’ displayed the most heterogeneous temporal response profiles among the cultivars tested. SPCSV infection followed an inverted parabolic pattern, achieving peak transcriptional dysregulation at 6 weeks (~2.8 log_10_ DEGs) before declining by 12 weeks. SPFMV demonstrated a consistent temporal decline. SPVD maintained constitutively low DEG counts throughout the infection course.

#### 3.3.5. GO Enrichment Analysis Reveals Systematic Perturbation of Cellular Homeostasis

Gene ontology overrepresentation analysis identified distinct functional landscapes between transcriptionally induced and repressed gene cohorts across virus–cultivar–timepoint combinations (Figure 7; Appendix A). Driver terms were prioritized from significant GO enrichments following g:SCS multiple testing correction (*p* < 0.05) to focus interpretation on the most reliable functional signatures. Transcriptionally suppressed genes exhibited pronounced enrichment for translational machinery components among driver terms. ‘Structural constituent of ribosome’ and ‘cytosolic ribosome’ ontologies were consistently overrepresented across multiple experimental conditions, particularly within ‘Tanzania’ genotypic backgrounds. Complementary driver term categories experiencing downregulation encompassed photosynthetic apparatus, RNA processing machinery, and carbohydrate transport systems.

Transcriptionally activated gene sets displayed robust enrichment for stress-responsive and regulatory functional modules among driver terms. These included ‘response to stimulus’ pathways, nucleic acid-binding activities, and transcriptional regulatory networks. Nuclear compartment-associated ontologies were predominantly enriched among upregulated gene cohorts. DNA-binding transcription factor activities and transcriptional regulator functions demonstrated consistent activation across experimental conditions.

Cultivar-dependent enrichment architectures emerged prominently among driver terms. ‘Tanzania’ manifested the most robust and temporally consistent functional enrichments across treatment modalities and sampling timepoints. Molecular function analysis revealed significant upregulation of oxidoreductase enzymatic activities and enzyme inhibitor functions among driver terms. The coordinated suppression of translational machinery concurrent with transcriptional regulator activation among the most statistically significant driver terms was observed across multiple conditions.

#### 3.3.6. Defense Pathway Differential Expression Reveals Virus- and Cultivar-Specific Activation Signatures

To understand how different defense mechanisms respond to viral infection across cultivars and timepoints, we analyzed the differential expression patterns of 11 literature-curated defense pathways using unscaled log_2_ fold change values relative to mock controls (Figure 8). We found that cell wall modification pathways showed the strongest upregulation (log_2_FC > 2) in specific SPCSV treatments, particularly in ‘New Kawogo’ and ‘Tanzania’ at early timepoints. NBS-LRR resistance pathways demonstrated moderate to strong upregulation (log_2_FC 1–2) in multiple conditions, with the strongest responses in ‘Tanzania’ SPCSV treatments and selected ‘New Kawogo’ conditions.

Protein quality control/HSP pathways exhibited pronounced upregulation in specific treatment combinations, with the strongest activation in certain SPCSV and SPVD treatments. RNA silencing pathways showed consistent moderate upregulation across diverse conditions, indicating broad antiviral defense activation. Hormone signaling pathways showed contrasting patterns: salicylic acid signaling exhibited moderate upregulation across multiple conditions, while jasmonic acid/ethylene pathways showed more restricted activation. Secondary metabolism pathways were predominantly downregulated across most conditions (log_2_FC < 0), consistent with metabolic redirection toward defense responses. Pattern recognition pathways showed selective upregulation in specific conditions, particularly in certain ‘Tanzania’ treatments, while miRNA-mediated regulation and RNA decay pathways displayed moderate activation levels across various treatment combinations, indicating post-transcriptional regulation of antiviral responses.

#### 3.3.7. Gene Expression in Viral Infections

The transcriptional response to SPCSV, SPFMV, and SPVD across the three sweet potato cultivars ‘Beauregard’, ‘Tanzania’, and ‘New Kawogo’ reveals significant genotype- and virus-specific differences. ‘Beauregard’ showed early and transient gene expression changes under SPCSV, a lack of response to SPFMV, and strong activity under SPVD at weeks 3 and 12, indicating susceptibility and stress-induced responses (Appendix A). In contrast, ‘Tanzania’ exhibited robust and sustained transcriptional activation, particularly in response to SPVD, with thousands of differentially expressed genes indicating active immune and metabolic engagement (Appendix A). ‘New Kawogo’ maintained a minimal response to SPVD, while showing modest, virus-specific changes under SPCSV and SPFMV, suggesting a degree of tolerance or resistance to these viruses (Appendix A; Table 1).

At week 3, ‘Beauregard’ displayed divergent transcriptional responses to SPCSV and SPVD, with downregulated genes affecting membrane systems and transcription, and upregulated genes pointing to an active redox and signaling response. ‘New Kawogo’ showed targeted suppression of intracellular transport and organelle activity under SPCSV, and a pronounced repression of chloroplast and redox functions under SPFMV. Conversely, ‘Tanzania’ broadly activated stress response, hormone signaling, and biosynthetic genes across all viruses, highlighting a resilient transcriptomic profile marked by organelle activity, redox catalysis, and transcriptional control (Appendix A).

By weeks 6 and 12, the contrast sharpened. ‘New Kawogo’ showed continued metabolic suppression under SPCSV, alongside the activation of detoxification genes. ‘Tanzania’ exhibited extensive repression of energy-related and structural pathways, coupled with selective reactivation of transcriptional and stress response systems under both SPCSV and SPVD, indicating a toggling between energy conservation and adaptive defense. At week 12, ‘Beauregard’ experienced broad transcriptional dysregulation under SPVD, with generalized upregulation lacking immune specificity signifying viral exploitation. In comparison, ‘Tanzania’ balanced partial suppression with coordinated upregulation of immune, detoxification, and organelle remodeling pathways, reinforcing its adaptive resistance (Appendix A).

#### 3.3.8. Potential Molecular Markers for Viral Resistance and Susceptibility

Analysis revealed molecular markers linked to viral resistance in ‘New Kawogo’ and susceptibility in ‘Beauregard’ sweet potato cultivars, with GO term classification highlighting candidate genes consistently up- or downregulated, while ‘Tanzania’ showed moderate marker expression.

##### Markers Associated with Moderate Tolerance in ‘New Kawogo’

‘New Kawogo’ displayed virus-specific, time-resolved immune responses. Against SPFMV, the cultivar employed a tolerance strategy marked by early immune activation and downregulation of metabolic pathways, followed by steady immune vigilance and eventual metabolic recovery. In contrast, SPCSV elicited a sustained, redox-regulated defense involving detoxification enzymes, suppression of viral exploitation pathways, and systemic immune responses. Minimal DEG activity during SPVD co-infection suggested that individual virus responses were sufficient to suppress synergistic disease effects. Key markers include sustained upregulation of RNA polymerase II-specific transcription factors, ATP-binding proteins, and redox enzymes such as monooxygenases. Notably, systemic acquired resistance (SAR) markers and glyoxylate cycle enzymes (e.g., malate synthase) were activated by 6 and 12 WPI, respectively, signifying a shift toward energy-efficient defense and recovery. Upregulated protein kinases, peroxisomal detox enzymes, and chromatin repair proteins further reflected robust immune reprogramming and transcriptional stability under prolonged stress. These findings demonstrate ‘New Kawogo’’s layered and adaptive immunity, offering promising molecular markers for breeding virus-resistant sweet potato varieties (Table 2; Appendix A).

##### Virus-Specific Immune Signatures in ‘Beauregard’

The sweetpotato cultivar ‘Beauregard’, highly susceptible to sweet potato virus disease (SPVD), exhibits delayed, disjointed, and ineffective transcriptional responses to viral infection, in stark contrast to resistant cultivars like ‘New Kawogo’. Across 3, 6, and 12 WPI, transcriptomic analysis reveals early immune suppression and viral manipulation, particularly in response to SPCSV, where crucial defense pathways are downregulated despite metabolic activation. SPVD co-infection further exacerbates this disruption, with widespread suppression of structural, hormonal, and metabolic defenses at early stages, followed by late-stage stress responses that are poorly coordinated and likely insufficient. Downregulated genes included DNA-binding transcription factors, glycosyltransferases, and hormone biosynthesis enzymes, particularly those involved in ethylene and abscisic acid (ABA) pathways. By 6 weeks post-SPVD infection, critical functions such as sucrose synthase activity and zinc ion binding were also suppressed, compromising sugar metabolism and enzymatic regulation. At 12 weeks, repression extended to monooxygenases, protein phosphatase inhibitors, and SUMOylation enzymes undermining stress signaling, protein regulation, and recovery potential.

The cultivar’s inability to mount a timely, targeted, and sustained resistance underscores its vulnerability and highlights the critical role of transcriptional regulation, hormone signaling, and metabolic homeostasis in effective viral resistance key pathways that can inform sweetpotato breeding for improved resilience (Table 3; Appendix A).

##### Markers Associated with Moderate Tolerance in ‘Tanzania’

The moderately tolerant sweetpotato cultivar ‘Tanzania’ exhibited early and sustained transcriptional repression in response to SPCSV and SPVD infections, with key defense and regulatory pathways such as DNA-binding transcription factors, glycosyltransferases, and ethylene biosynthesis suppressed by 3 WPI. This repression extended to sucrose metabolism, ATP- and metal ion-binding functions, and energy regulation by 6 WPI, indicating disrupted stress signaling and enzymatic activity. Although some stress-related pathways, like MAPK signaling, were activated, their response was uncoordinated. By 12 WPI, immune modulation and redox balance further deteriorated due to the downregulation of monooxygenases, SUMOylation enzymes, and ABA signaling components. While late-stage compensatory responses involving photosystem adjustment, chromatin remodeling, and small GTPase signaling emerged, they appeared fragmented and inadequate to counteract the earlier immune suppression (Table 4; Appendix A).

##### Comparative Transcriptomic Analysis of ‘Beauregard’ and ‘New Kawogo’

Comparative transcriptomic analysis between the susceptible ‘Beauregard’ and highly tolerant ‘New Kawogo’ sweetpotato cultivars reveals distinct differences in the timing, coordination, and effectiveness of their responses to SPCSV and SPVD infections. ‘Beauregard’ exhibited early widespread suppression of critical functions, reflecting a stress-induced ‘panic’ response, followed by limited mid-phase recovery and inadequate late-stage reactivation, particularly under SPVD. In contrast, ‘New Kawogo’ employed a regulated, energy-conserving early response, maintained immune–metabolic balance during the mid-phase, and launched a robust recovery by 12 weeks, marked by transcriptional reprogramming. These findings highlight that precise regulation and timely immune activation are key determinants of viral tolerance in sweetpotato (Table 5; Appendix A).

##### Comparative Molecular Markers Underpinning Resistance and Susceptibility in Sweet Potato Cultivars

To elucidate the molecular basis of sweetpotato resistance and susceptibility to SPVD and SPCSV, candidate gene markers were identified from the tolerant ‘New Kawogo’ and susceptible ‘Beauregard’ cultivars based on consistent transcriptional patterns and GO term annotations across infection stages. ‘New Kawogo’ exhibited early and sustained activation of transcription factors, ATP-binding proteins, redox enzymes, and energy-regulating pathways, including RNA polymerase II factors, SAR markers, and glyoxylate cycle enzymes indicating a coordinated, energy-efficient immune response and recovery. In contrast, ‘Beauregard’ showed early repression of key defense-related genes, including transcription factors and hormone biosynthesis enzymes, with continued downregulation of metabolic, redox, and protein regulatory pathways through 12 WPI, reflecting delayed and disorganized responses that compromise immunity and recovery (Table 3; Appendix A).

The moderately tolerant ‘Tanzania’ also displayed early repression of key regulatory pathways, including sugar metabolism and stress signaling, with only partial and disorganized late-stage compensatory responses. Overall, resistance was associated with timely and robust gene activation, while susceptibility correlated with early repression and impaired recovery mechanisms (Table 4; Appendix A).

## 4. Discussion

Sweetpotato is a vital food security crop across tropical and subtropical regions, yet it suffers significant yield losses from sweet potato virus disease (SPVD), a synergistic infection caused by sweet potato chlorotic stunt virus (SPCSV) and sweet potato feathery mottle virus (SPFMV). This study offers a multi-dimensional analysis of how the three cultivars ‘New Kawogo’, ‘Tanzania’, and ‘Beauregard’ respond phenotypically and transcriptionally to single and co-infections over a 12-week period. By integrating phenotypic assessments with transcriptomic profiling and gene ontology (GO) enrichment, we revealed cultivar-specific defense strategies and candidate molecular markers that differentiate resistance, tolerance, and susceptibility.

Phenotypic symptom severity closely aligned with transcriptomic responses across the cultivars. ‘New Kawogo’ showed minimal symptoms and full recovery from SPVD by 12 WPI, indicative of strong intrinsic resistance [41,42]. This finding aligns with previous observations in other tolerant landraces [43], exhibiting early and coordinated transcriptional defense activation. In contrast, ‘Beauregard’ displayed severe and persistent symptoms, particularly under SPVD co-infection. This phenotype correlated with transcriptional disarray and poor immune regulation, consistent with patterns observed in highly susceptible genotypes. ‘Tanzania’ presented moderate symptoms and eventual recovery, suggesting partial tolerance [38,44] as observed in fields under high virus pressure [45].

This study investigated the role of virus-derived small interfering RNAs (vsiRNAs) in the antiviral defense of sweetpotato against SPFMV and SPCSV in the ‘New Kawogo’ and ‘Tanzania’ cultivars, building on the work by Bednarek et al., 2021 [15].

We found that RNA silencing is a critical weapon in sweetpotatoes’ antiviral response. ‘Tanzania’ stood out with robust vsiRNA production against SPCSV compared to ‘New Kawogo’, suggesting a robust antiviral response yielding to a stronger natural defense that makes it one of the top candidates for breeding virus-resistant varieties [46,47]. In contrast, ‘New Kawogo’ shows weaker RNA silencing, potentially leaving it more vulnerable [48]. These differences, alongside insights from ‘Beauregard’ [15], hint that some sweetpotato cultivars are naturally better at fighting viruses. This highlights how cultivar-specific responses can guide breeding programs to develop virus-resistant sweetpotatoes [46,47,49,50].

The study also pinpoints a critical window around 6 WPI when vsiRNA activity peaks, signaling a high point in the plant’s defense or viral activity. By 12 weeks, vsiRNA levels often drop, suggesting the plant may be gaining control or the virus is slowing down, calling for longer-term studies to optimize intervention timing. ‘Beauregard’ shows similar patterns, reinforcing the broad relevance of these findings [15,51].

SPCSV elicited higher vsiRNA levels than SPFMV, likely due to its silencing suppressor, enhancing SPFMV in SPVD, which suggests it is less of a target for the plant’s defenses [51] and highlighting its dominant role in SPVD synergy [52]. This shows that there is a need for strategies focused on neutralizing SPCSV’s suppressor. SPFMV, by contrast, triggers less of a response. In SPVD infections, SPCSV boosts SPFMV’s impact [52]. To tackle SPVD, strategies should focus on neutralizing SPCSV’s silencing suppressor, possibly through targeted RNAi tools. SPFMV, while less aggressive, still plays a role in co-infections, so SPVD-specific approaches are vital, especially in heavily affected regions. These insights into RNA silencing offer a roadmap for breeding resilient sweet potato varieties and designing precise, effective ways to manage viral diseases.

This response is comparable to genotypes with intermediate resistance that rely on delayed yet sustained activation of immune pathways. Principal component analysis (PCA) and hierarchical clustering of differentially expressed genes (DEGs) underscored the role of host genotype and infection timepoint as the dominant factors shaping transcriptional responses, outweighing virus-specific effects.

‘New Kawogo’ demonstrated a rapid, targeted, and resource-efficient immune strategy. At 3 WPI, it upregulated ATP-binding proteins, transcription factors, and redox enzymes, while downregulating primary metabolic processes, indicating a strategic reallocation of energy toward immune function. By 6 WPI, genes involved in detoxification and kinase signaling were activated, and by 12 WPI, the expression profile transitioned to include DNA repair genes and those associated with systemic acquired resistance (SAR) [10,43]. Interestingly, its response to SPVD co-infection was relatively muted, suggesting a strong basal resistance to both SPCSV and SPFMV that prevents the synergistic amplification typically associated with SPVD [53]. The consistent expression of oxidoreductases, hormone-responsive transcription factors, and kinases highlights ‘New Kawogo’ as an excellent candidate for resistance breeding in virus-endemic areas.

‘Tanzania’ exhibited a flexible, adaptive response that evolved with the progression of infection. In the early stages, broad activation of stress response genes, salicylic acid (SA) and abscisic acid (ABA) signaling [54], and secondary metabolism was observed [49,55,56]. By 6 WPI, the cultivar downregulated energy-consuming pathways such as the TCA cycle while upregulating genes related to protein quality control and DNA repair [57]. At 12 WPI, sustained induction of MAPK signaling and detoxification enzymes indicated ongoing immune engagement [58,59,60]. While basal metabolic and hormonal signaling pathways were repressed, this appeared to be a strategic reprioritization of cellular functions rather than a sign of immune collapse [58]. These findings suggest that the partial tolerance of ‘Tanzania’ is driven by its capacity to fine-tune immune responses across time.

In contrast, ‘Beauregard’ mounted a disorganized and largely ineffective immune response. As early as 3 WPI, downregulation of stress signaling components, ATPases, and key transcription factors suggested either viral suppression or a compromised ability to perceive infection. SPVD exposure led to pronounced repression of the ethylene and jasmonic acid hormone pathways, both of which are critical for antiviral defense [15,61]. Upregulation of metabolic enzymes in ‘Beauregard’ appeared to be a stress response rather than a deliberate immune strategy, as is often seen in abiotic stress responses like the drought response [62]. By 6 and 12 WPI, transcriptional profiles failed to show recovery, with only sporadic upregulation of transporters and hydrolases. This fragmented response underscores the cultivar’s vulnerability to immune suppression and redox imbalance [15].

Cross-cultivar comparisons revealed distinct molecular defense signatures. ‘New Kawogo’ consistently upregulated regulatory genes such as RNA polymerase II subunits, oxidoreductases, and glyoxylate cycle enzymes, while suppressing host processes exploited by viruses. ‘Tanzania’ adapted by suppressing energy-intensive metabolism and shifting toward late-stage detoxification and immune signaling [63]. In contrast, ‘Beauregard’ showed early suppression of critical immune and metabolic regulators and failed to recover by later stages, confirming findings by Bednarek et al., 2021 [15] GO enrichment analysis indicated that tolerant cultivars favored the activation of stress responses, redox regulation, and nuclear processes, while susceptible cultivars exhibited repression of ribosomal and photosynthetic pathways. Defense-related pathways including NBS-LRR signaling, RNA silencing, and hormonal regulation were consistently upregulated in ‘New Kawogo’ [58,64] and to a lesser extent in ‘Tanzania’, but remained inactive in ‘Beauregard’.

Comparative transcriptomic profiling identified key molecular markers underpinning these resistance phenotypes. In ‘New Kawogo’, early and sustained expression of genes associated with RNA polymerase II regulation, redox balance, and SAR signaling illustrated a well-orchestrated immune response [10,65]. ‘Beauregard’ exhibited repression of genes associated with hormone synthesis, transcriptional regulation, and defense signaling, consistent with delayed and ineffective immunity [15]. ‘Tanzania’ displayed a hybrid profile initial repression of sugar metabolism and early stress responses, followed by disorganized activation of detoxification and signaling genes. These patterns emphasize that resistance depends on timely and robust activation of immunity, while susceptibility is characterized by early immune silencing and transcriptional instability.

Temporal DEG and UpSet plot analyses further supported these observations. SPVD co-infection elicited the most dramatic transcriptional shifts, with ‘Tanzania’ exhibiting the highest number of DEGs and suggesting active immune reprogramming. Across all treatments, certain genes emerged as promising molecular markers, including ATP-binding proteins, oxidoreductases, SUMOylation enzymes, and glycosyltransferases [59]. These genes represent valuable targets for marker-assisted selection and genome editing to develop virus-resilient sweet potato lines.

This study provides a comprehensive foundation for improving sweetpotato virus resistance by revealing genotype-specific defense mechanisms and identifying molecular markers for breeding. The phased and cost-effective immune response in ‘New Kawogo’ makes it a model for resistance breeding. The transcriptional adaptability of ‘Tanzania’ provides insights into stress resilience strategies. Conversely, the immune disorganization of ‘Beauregard’ offers critical lessons on the consequences of viral suppression. Moving forward, resistance breeding should combine phenotypic and molecular datasets to identify key defense-associated QTLs, regulatory networks, and gene modules. Expanding the analysis to a broader range of genotypes and incorporating additional layers such as epigenomic, proteomic, and metabolomic data will enable a deeper understanding of the multi-omic foundations of viral resistance. Functional validation of candidate genes using genome editing tools like CRISPR/Cas will accelerate the development of durable, virus-resistant sweet potato cultivars.

## 5. Conclusions

This study underscores the distinct molecular defense strategies employed by sweetpotato cultivars in response to viral infections, particularly SPVD. The highly tolerant cultivar ‘New Kawogo’ demonstrated a phased and targeted immune response characterized by the early activation of defense-related genes, suppression of energy-intensive processes, and transcriptional stability across infection stages. Rather than mounting a broad-spectrum gene activation, its resistance relied on precise temporal coordination, redox regulation, and strategic engagement of systemic acquired resistance pathways. In contrast, ‘Beauregard’ exhibited a fragmented and delayed response marked by widespread suppression of critical immune and metabolic functions, reflecting susceptibility to viral interference and an inability to restore homeostasis. ‘Tanzania’ presented an intermediate, adaptive strategy, balancing suppression and activation of immune pathways over time.

These findings highlight key regulatory genes and temporal expression patterns that can serve as molecular markers for breeding virus-resilient sweetpotato cultivars. However, the use of a diploid reference genome for a hexaploid species, along with limited sampling timepoints and a lack of environmental variability, points to the need for expanded genomic resources and functional validation of candidate genes. Despite these limitations, the integrated transcriptomic and phenotypic insights provided here offer a valuable framework for precision breeding and the development of robust, virus-tolerant cultivars suited for deployment in SPVD-endemic regions.

## Figures and Tables

**Figure 1 biology-14-01541-f001:**
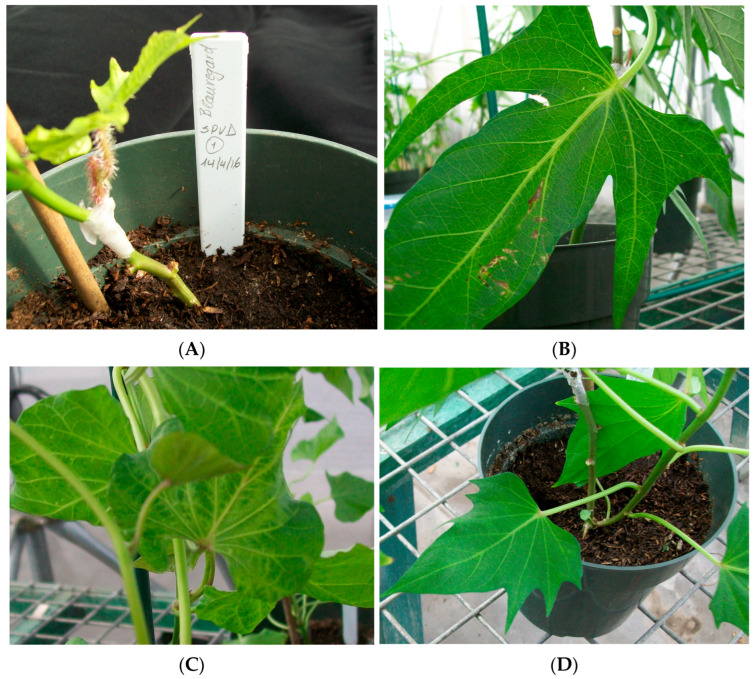
Symptoms observed on leaves of plants co-infected with SPCSV and SPFMV at different timepoints. (**A**) Graft inoculation using *I. setosa*. (**B**) ‘Tanzania’ at 6 WPI showing mild symptoms of veinal chlorosis. (**C**) ‘Beauregard’ at 6 WPI showing vein clearing, mosaic, and roll down. (**D**) ‘New Kawogo’ at 6 WPI showing no symptoms.

**Figure 2 biology-14-01541-f002:**
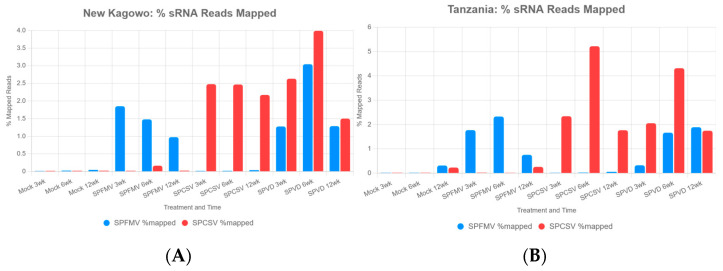
Bar charts illustrate the mean percentage of virus-derived small interfering RNA (vsiRNA) reads mapped to SPFMV and SPCSV genomes in (**A**) ‘New Kawogo’ and (**B**) ‘Tanzania’ across mock, SPFMV, SPCSV, and SPVD treatments at 3, 6, and 12 WPI.

**Figure 3 biology-14-01541-f003:**
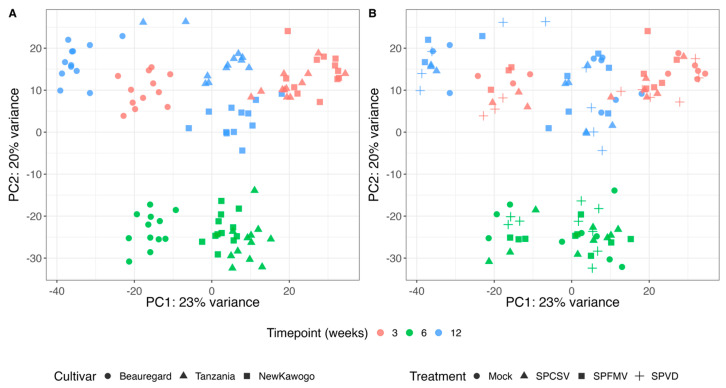
Principal component analysis of sweet potato transcriptional responses to viral infection. (**A**) Cultivar and temporal clustering. Samples are colored by timepoint (red: 3 weeks; green: 6 weeks; blue: 12 weeks) and shaped by cultivar (circle: ‘Beauregard’; triangle: ‘Tanzania’; square: ‘New Kawogo’). (**B**) Treatment and temporal clustering. Samples are colored by timepoint and shaped by treatment (circle: mock; triangle: SPCSV; square: SPFMV; cross: SPVD). Principal components explain 44% of total variance (PC1: 23%; PC2: 21%).

**Figure 4 biology-14-01541-f004:**
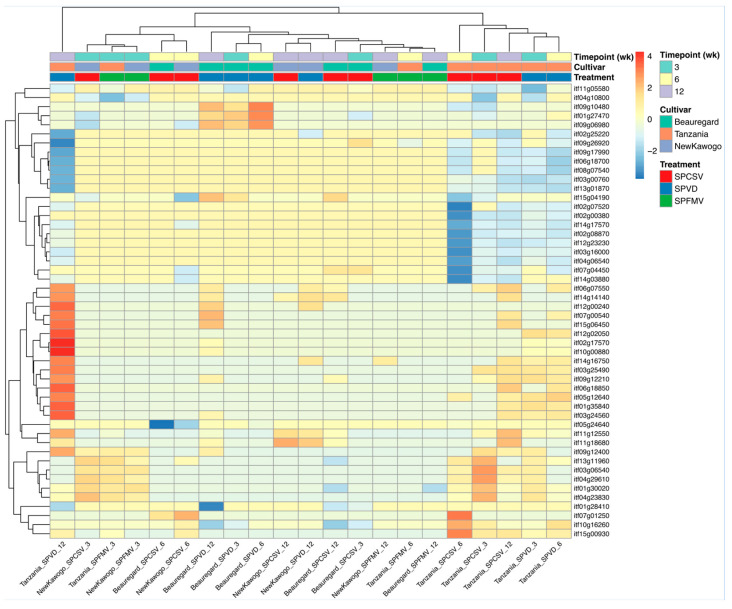
Expression patterns of top 50 differentially expressed genes. Heatmap shows row-scaled log_2_ fold change values for the 50 most significantly differentially expressed genes (lowest adjusted *p*-values) across treatment comparisons. Columns represent individual comparisons organized by cultivar (‘Beauregard’, ‘Tanzania’, ‘New Kawogo’), treatment (SPCSV, SPFMV, SPVD), and timepoint (3, 6, 12 weeks). Hierarchical clustering was performed for both genes (rows) and comparisons (columns). Values represent z-scored log_2_ fold changes relative to control conditions.

**Figure 5 biology-14-01541-f005:**
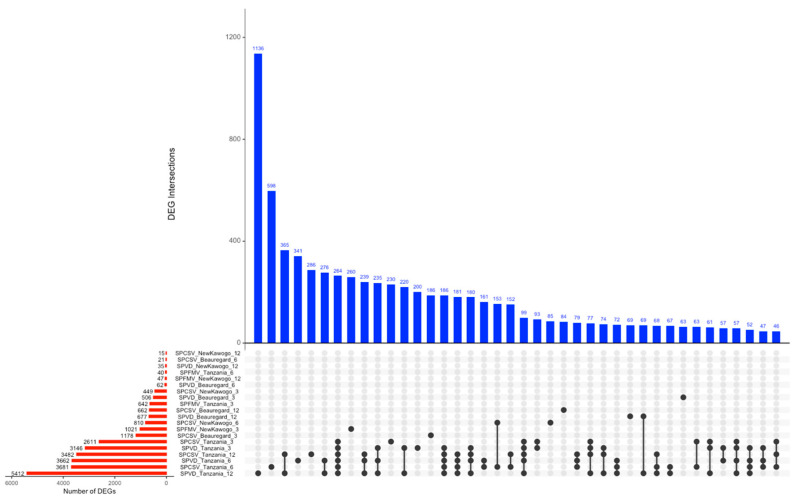
UpSet plot of DEG intersections across treatment conditions. (**Bottom panel**) Set sizes (number of DEGs) for each condition labeled as Virus_Cultivar_Timepoint. (**Top panel**) Intersection sizes between condition combinations, with connected dots indicating which sets contribute to each intersection. Sets are ordered by frequency, with intersections showing both shared and condition-specific differential expression signatures across viral treatments, cultivars, and timepoints.

**Figure 6 biology-14-01541-f006:**
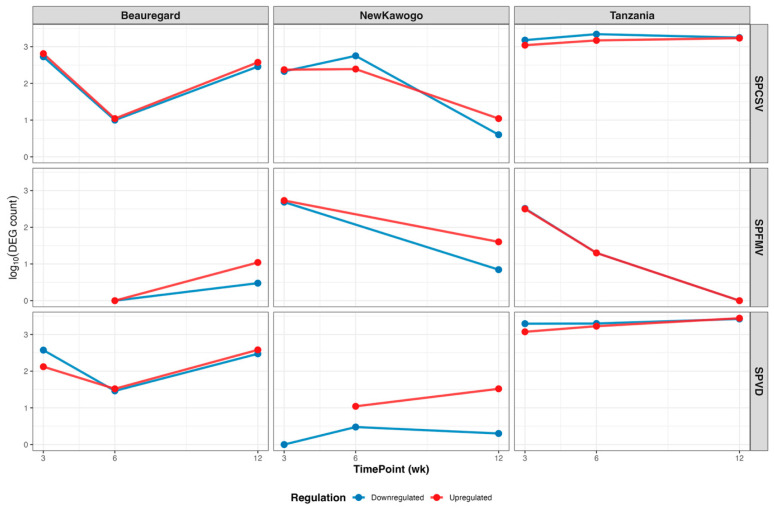
Temporal dynamics of differential gene expression across virus–cultivar combinations. Line plots show log10-transformed DEG counts over time (3, 6, 12 weeks) for each virus–cultivar combination. Red lines indicate upregulated genes (log_2_FC ≥ 0.5); blue lines indicate downregulated genes (log_2_FC ≤ −0.5). Panels are organized by virus treatment (rows) and cultivar (columns). DEGs were filtered using padj < 0.05 and |log_2_FC| ≥ 0.5 thresholds.

**Figure 7 biology-14-01541-f007:**
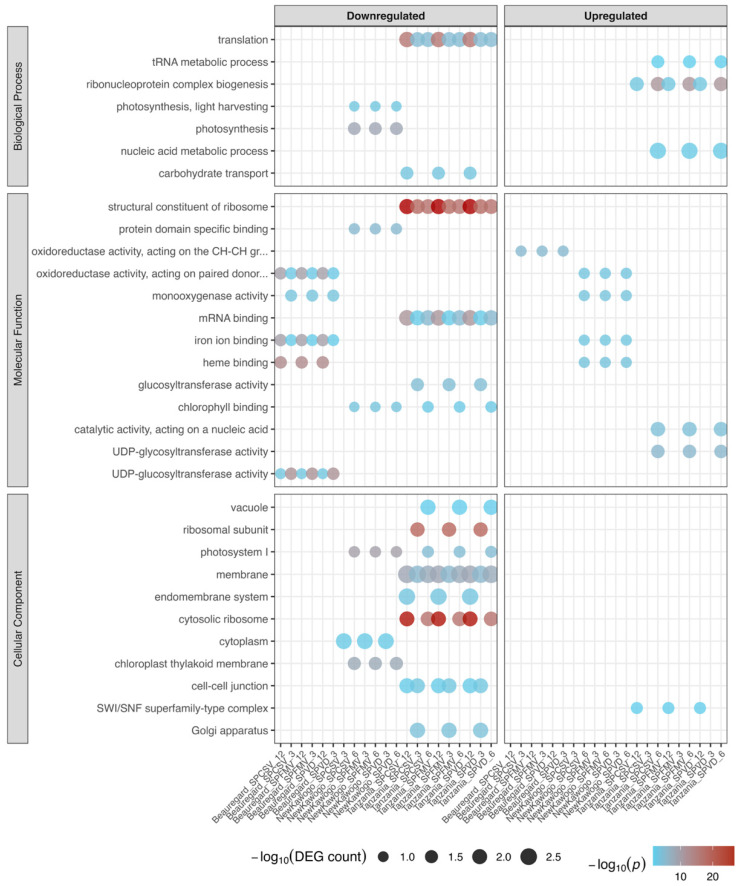
Functional enrichment analysis of differentially expressed genes across treatment conditions. Dot plots show significantly enriched gene ontology driver terms (*p* < 0.05, g:SCS correction) for downregulated (**left**) and upregulated (**right**) genes within each condition. Driver terms represent the most statistically robust and biologically relevant enrichments. Terms are organized by GO domain: biological process, molecular function, and cellular component. Dot size represents DEG count contributing to enrichment; dot color represents −log10(*p*-value). Conditions are labeled as Cultivar_Virus_Timepoint.

**Figure 8 biology-14-01541-f008:**
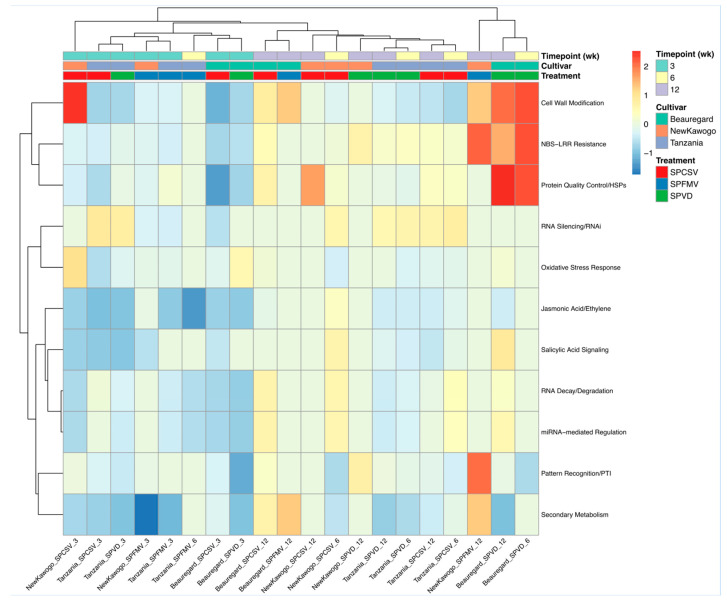
Defense pathway differential expression across virus–cultivar–timepoint combinations. Heatmap shows unscaled log_2_ fold change values for defense pathway gene sets relative to mock controls. Red indicates upregulation; blue indicates downregulation. Pathways include cell wall modification, NBS-LRR resistance, protein quality control/HSPs, RNA silencing, oxidative stress response, hormone signaling (SA and JA/ET), RNA decay, miRNA regulation, pattern recognition, and secondary metabolism. Conditions are labeled as Cultivar_Virus_Timepoint and organized by cultivar (‘Beauregard’: red; ‘New Kawogo’: orange; ‘Tanzania’: blue), treatment (SPCSV: red; SPFMV: blue; SPVD: green), and timepoint (3, 6, 12 weeks).

**Table 1 biology-14-01541-t001:** Time-resolved gene expression response to SPCSV, SPFMV, and SPVD in each cultivar (‘Beauregard’, ‘Tanzania’, and ‘New Kawogo’) with absolute log_2_ fold change ≥ 1.

Virus	Time (Week)	Cultivar	Downregulated Genes	Upregulated Genes
SPCSV	3	‘Beauregard’	68	99
SPCSV	3	‘Tanzania’	149	82
SPCSV	3	‘New Kawogo’	33	37
SPCSV	6	‘Beauregard’	5	5
SPCSV	6	‘Tanzania’	514	168
SPCSV	6	‘New Kawogo’	108	40
SPCSV	12	‘Beauregard’	34	37
SPCSV	12	‘Tanzania’	120	42
SPCSV	12	‘New Kawogo’	1	9
SPFMV	3	‘Beauregard’	0	0
SPFMV	3	‘Tanzania’	13	6
SPFMV	3	‘New Kawogo’	42	52
SPFMV	6	‘Beauregard’	0	0
SPFMV	6	‘Tanzania’	0	1
SPFMV	6	‘New Kawogo’	0	0
SPFMV	12	‘Beauregard’	0	0
SPFMV	12	‘Tanzania’	0	0
SPFMV	12	‘New Kawogo’	0	0
SPVD	3	‘Beauregard’	93	31
SPVD	3	‘Tanzania’	277	43
SPVD	3	‘New Kawogo’	1	0
SPVD	6	‘Beauregard’	9	16
SPVD	6	‘Tanzania’	178	89
SPVD	6	‘New Kawogo’	1	0
SPVD	12	‘Beauregard’	52	73
SPVD	12	‘Tanzania’	659	577
SPVD	12	‘New Kawogo’	0	2

**Table 2 biology-14-01541-t002:** Potential molecular resistance markers in ‘New Kawogo’.

#	Gene Function/GO Term	Timepoint	Pathogen	Resistance Function	Gene
1	DNA-binding TFs (RNA Pol II-specific)	3–12 WPI	SPFMV/SPCSV	Sustained transcriptional activation of immune genes	itf02g15710
2	ATP-binding proteins	3–12 WPI	SPFMV/SPCSV	Energy-efficient signaling and stress response	itf12g25640, itf12g25640, itf05g04270
3	mRNA-binding proteins	3 WPI	SPFMV/SPCSV	Post-transcriptional regulation and immune priming	itf03g16440, itf09g14950
4	Monooxygenases/oxidoreductases	6 WPI	SPFMV/SPCSV	ROS detoxification and redox balance	itf06g16700
5	Systemic acquired resistance (SAR) markers	6 WPI	SPFMV/SPCSV	Long-range immune signaling	itf01g07140, itf12g22230
6	Glyoxylate cycle enzymes (e.g., malate synthase)	12 WPI	SPFMV/SPCSV	Energy metabolism adaptation and recovery	itf07g01900
7	Protein kinases (MAPK, LRR-like)	6–12 WPI	SPFMV/SPCSV	Immune signal transduction	itf03g07330
8	Photoreceptor-related transcription factors	12 WPI	SPFMV/SPCSV	Environmental sensing and late-phase regulation	itf04g14470
9	Peroxisomal enzymes (e.g., catalase)	6 WPI	SPFMV/SPCSV	ROS detoxification	itf12g02300
10	DNA repair and chromatin regulators	12 WPI	SPFMV/SPCSV	Genome integrity during stress recovery	itf02g21900

**Table 3 biology-14-01541-t003:** Molecular markers differentiating resistance and susceptibility.

#	Gene Function/GO Term	Timepoint	Pathogen	Susceptibility Indicator	Gene
1	DNA-binding transcription factors	3 and 12 WPI	SPCSV/SPVD	Downregulated: impairs immune gene activation	itf02g15710
2	Glycosyltransferase activity	3, 6, 12 WPI	SPVD	Suppressed: limits structural defenses and metabolite biosynthesis	itf00g21310, itf06g06410
3	1-aminocyclopropane-1-carboxylate synthase	3 WPI	SPVD	Suppressed: disrupts ethylene biosynthesis and stress signaling	itf12g18840
4	Abscisic acid (ABA) binding and signaling	12 WPI	SPVD	Downregulated: weakens hormonal regulation of stress responses	itf12g11260
5	ATP-binding and ATP hydrolysis activity	3 WPI	SPCSV	Suppressed: impairs energy management	itf12g25640
6	SUMOylation enzymes	12 WPI	SPVD	Repression: limits protein regulation under stress	itf08g08490
7	Metal ion (iron, zinc, heme) binding	3–12 WPI	SPCSV/SPVD	Suppressed: disrupts enzymatic activity and redox balance	itf03g17170, itf13g04600
8	Sucrose synthase activity	6 WPI	SPVD	Downregulated: disrupts sugar metabolism and energy provisioning	itf02g04900
9	Monooxygenases/oxidoreductases	12 WPI	SPVD	Suppressed: limits ROS detoxification and immune signaling	
10	Protein phosphatase inhibitors	12 WPI	SPVD	Repressed: affects immune signal modulation	itf12g11260, itf15g22590

**Table 4 biology-14-01541-t004:** Moderately tolerant markers, ‘Tanzania’.

#	Gene Function/GO Term	Timepoint	Pathogen	Susceptibility Indicator	Gene
1	DNA-binding transcription factors	3 and 12 WPI	SPCSV/SPVD	Downregulated: impairs immune gene activation	itf02g15710
2	Glycosyltransferase activity	3, 6, 12 WPI	SPVD	Suppressed: limits structural defenses and metabolite biosynthesis	itf00g21310, itf06g06410
3	1-aminocyclopropane-1-carboxylate synthase	3 WPI	SPVD	Suppressed: disrupts ethylene biosynthesis and stress signaling	itf12g18840
4	Abscisic acid (ABA) binding and signaling	12 WPI	SPVD	Downregulated: weakens hormonal regulation of stress responses	itf12g11260
5	ATP-binding and ATP hydrolysis activity	3 WPI	SPCSV	Suppressed: impairs energy management	itf12g25640
6	SUMOylation enzymes	12 WPI	SPVD	Repressed: limits protein regulation under stress	itf08g08490
7	Metal ion (iron, zinc, heme) binding	3–12 WPI	SPCSV/SPVD	Suppressed: disrupts enzymatic activity and redox balance	itf03g17170, itf13g04600
8	Sucrose synthase activity	6 WPI	SPVD	Downregulated: disrupts sugar metabolism and energy provisioning	itf02g04900
9	Monooxygenases/oxidoreductases	12 WPI	SPVD	Suppressed: limits ROS detoxification and immune signaling	itf06g16700
10	Protein phosphatase inhibitors	12 WPI	SPVD	Repressed: affects immune signal modulation	itf12g11260, itf15g22590
11	Ubiquitin-mediated protein degradation	3 WPI	SPVD	Downregulated: limits protein turnover and stress response	itf01g33310
12	Photosystem stoichiometry adjustment	12 WPI	SPVD	Upregulated; compensates energy conversion imbalance	itf03g10900
13	Chromatin remodeling and gene regulation	12 WPI	SPVD	Upregulated; enhances transcriptional flexibility under stress	itf05g18650
14	Polycomb repressive complex activity	12 WPI	SPVD	Upregulated: regulates chromatin structure and gene silencing	itf14g19980
15	MAPK signaling cascade	6 WPI	SPVD	Upregulated: facilitates stress signaling and response	itf12g20570
16	Small GTPase activation	12 WPI	SPVD	Upregulated: promotes cellular signaling and vesicle trafficking	itf09g25080

**Table 5 biology-14-01541-t005:** Transcriptomic analysis of ‘Beauregard’ and ‘New Kawogo’.

Functional Category	‘New Kawogo’ (Resistant)	‘Beauregard’ (Susceptible)
Early Defense (3 WPI)	Selective suppression of lipids; ATP-binding and hypoxia response activated	Suppression of cell wall, lipid metabolism, and hormone signaling (panic)
Mid-Phase Response (6 WPI)	ROS detoxification, photosynthesis, SAR markers upregulated	Minimal recovery; SPVD suppresses growth and hormone pathways
Recovery Phase (12 WPI)	Full activation of DNA repair, glyoxylate cycle, mRNA surveillance	Delayed activation; chronic suppression of ABA, SUMO, sugar transport
Transcriptional Regulation	Early TFs and mRNA-binding proteins active	Late TF response; SUMOylation enzymes suppressed
Hormone Signaling	Maintains ABA and hypoxia responses, photoreceptors at recovery stag	ABA, JA, and Ca^2+^ signaling downregulated (especially under SPVD)
Energy Management	Glyoxylate/TCA cycle activated for efficient recovery	Sucrose metabolism genes suppressed
Redox and Detoxification	Monooxygenases and peroxisomal enzymes timely upregulated	Redox response delayed and incomplete
Transport and Structural Repair	Transporters, DNA repair, and membrane proteins reactivated by 12 WPI	Suppressed transporters, protein folding and phosphatase inhibitors

## Data Availability

Restrictions apply to the availability of these data. Data was obtained from the Genomic Tools for Sweet Potato Improvement (GT4SP) project, funded by the Bill & Melinda Gates Foundation (Contract ID: OPP1052983), and is available from the corresponding author with the permission of the funding body.

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
