# Peer review of "Virus-Specific Defense Responses in Sweetpotato: Transcriptomic Insights into Resistance and Susceptibility to SPFMV, SPCSV, and SPVD"

_biology, 2025, doi:10.3390/biology14111541_

Round 1

Reviewer 1 Report

Comments and Suggestions for Authors

The manuscript entitled “Virus-specific defense responses in Sweetpotato: Transcriptomic insights into resistance and susceptibility to SPFMV, SPCSV and SPVD” describes the transcriptomic changes occurring in virus susceptible, tolerant and moderately tolerant sweet potato varieties. The authors have worked hard (commendable effort) in stitching an excellent scientific story which would be of significant use for a broad range of authors. The data analysis and presentation are perfect. However, I have a few questions related to experimental design of this study.

  1. The authors have not stated the scientific rationale behind the time points that they have chosen to analyze. It would have been excellent is authors could have shown the disease progression in infected plant (or literature support) to justify the time points.
  2. I fail to understand as to why virus infected scions of an indicator host were used for grafting. This strategy would be fit for studying mobile signals of immune response not for studying the inherent immune response of the varieties under study. Since you are taking samples (for transcriptomic analyses) from scion and not the varieties themselves, so how would you explain what the inherent mechanisms are employed by varieties under study. It would have been excellent if the authors could have used another virus inoculation method (e.g infectious clones, insect feeding or mechanical inoculation). In case, the choice of virus inoculation method is only grafting then it would be better to use infected setosa as rootstock.
  3. The virus symptoms presented in figure 1 (A: vein clearing) are not obvious, similarly in B panel roll down is not clearly visible. I understand that typical symptoms may be hard to observe at certain times, but you can always show a control plant along with infected plant (which in your case is clearly missing).
  4. I did not see any data on virus detection in the samples used for analysis (correct me if I am wrong). It is important to show the virus presence in each sample at each time point (would be much better if you can show absolute quantification of virus at each sampling point, this would also help in understanding a critical virus load dependent relationship).

Author Response

Reviewer 1

Comments 1: The authors have not stated the scientific rationale behind the time-points that they have chosen to analyze. It would have been excellent if authors could have shown the disease progression in infected plant (or literature support) to justify the time points

Response 1: This study is building on previous work by Bednarek et al. 2021 and the time points were derived from this work. The three time points were meant to capture early, middle and late responses of the different cultivars.

Comment 2: I fail to understand as to why virus infected scions of an indicator host were used for grafting. This strategy would be fit for studying mobile signals of immune response not for studying the inherent immune response of the varieties under study. Since you are taking samples (for transcriptomic analyses) from scion and not the varieties themselves, so how would you explain what the inherent mechanisms are employed by varieties under study. It would have been excellent if the authors could have used another virus inoculation method (e.g infectious clones, insect feeding or mechanical inoculation). In case, the choice of virus inoculation method is only grafting then it would be better to use infected setosa as rootstock.

Response 2: Ipomoea setosa is a wild relative of sweetpotato that is very susceptible to sweetpotato viruses. It has therefore been used to either test for or inoculate sweetpotato viruses. Due it it’s susceptibility, it is used to increase viral loads for inoculation, that ensures that infection is achieved during experiments. It is easier to achieve this than other infection methods.

Comment 3: The virus symptoms presented in figure 1 (A: vein clearing) are not obvious, similarly in B panel roll down is not clearly visible. I understand that typical symptoms may be hard to observe at certain times, but you can always show a control plant along with infected plant (which in your case is clearly missing).

Response 3: This comment has been addressed. Photos to represent symptoms have also been changed. However, we showed one control plant (Beauregard) that had no symptoms.

Comment 4: I did not see any data on virus detection in the samples used for analysis (correct me if I am wrong). It is important to show the virus presence in each sample at each time point (would be much better if you can show absolute quantification of virus at each sampling point, this would also help in understanding a critical virus load dependent relationship).

Response 4: In reference to viral loads, we do agree with the reviewer.As previously mentioned, this work is building on to work by Bednarek et al., 2021 were analysis for viral load was only Beauregard has been done and published. Viral loads for the rest of the cultivars is underway, and we intend to publish it in another paper.

5. Additional clarifications

Due to quality if English, the whole manuscript was revised which cause a change in page numbering and paragraph arrangement.

Reviewer 2 Report

Comments and Suggestions for Authors

Abstract

Line 28: Sweetpotato or Sweet potato? Consistancy throughout the manuscript

Lines 31-32 and else where in the manuscript : I think brakets are not so much important. Same as at line 264.

Lines 43 : this looks like a repetition already found in the previous sentences

Introduction

Lines 50-53 : Too long sentence, please split it in two sentences

Line 54 : The sentence looks grammaticaly incorrect regarding the backdrop

Line 84. Sweetpotato virus disease (SPVD) =  (SPVD)

Line 100 and else where : SPCSV, SPFMV and SPVD infection = SPCSV and SPFMV infection or SPVD complex infection

Materials and methods

Line 121 : Which of the treatment was carried out by the I. setosa plants and how were others treatment carried out ?

Line 129 contradicts line 121. How many treatments were used in the study and what are they ?

Results

Lines 254 – 262 : results presented here look more general, I think readers may need to know more about the

Lines 268-269 : This not true. By observing the figure 2B, the mock control does not clearly show a distinct cluster from viral treatments.

Discussion

Line 570 and else where throughout the manuscript : full name should appear just once in core text of the manuscript (weeks post-infection WPI = WPI)

Line 573 : reference after ‘’defense activation’’

Line 694-696 : The sentence looks grammaticaly incorrect ; please check

References

Lines 705-708 : remove capital letters within sentences and keep constancy with others references, same at lines 717-718, 818-815 

Comments on the Quality of English Language

The english should be improved

Author Response

Comment 1: Line 28: Sweetpotato or Sweet potato? Consistency throughout the manuscript

Response 1: Sweetpotato as a crop is spelt without a space, however when talking about the viruses, there is space between the word “sweetpotato” as it is a name of a virus not the crop. Consistency has been checked thought the manuscript

Comment 2: Lines 31-32 and elsewhere in the manuscript: I think brackets are not so much important. Same as at line 264.

Response 2: Authors have addressed the comment made by the reviewer

Comment 3: Lines 43: this looks like a repetition already found in the previous sentences

Response 3: Authors do agree with the reviewer and have addressed the comment.

Introduction

Comment 4: Lines 50-53: Too long sentence, please split it in two sentences

Response 4: Authors agree with the reviewer and comment has been addressed by authors

Comment 5: Line 54: The sentence looks grammatically incorrect regarding the backdrop

Response 5: Sentence has been revised to address reviewer’s comment

Comment 6: Line 84. Sweetpotato virus disease (SPVD) = (SPVD)

Response 6: Spelling has been corrected to Sweet potato virus disease

Comment 7: Line 100 and elsewhere: SPCSV, SPFMV and SPVD infection = SPCSV and SPFMV infection or SPVD complex infection

Response 7: SPCSV= Sweet potato chlorotic stunt virus, SPFMV= Sweet potato feathery mottle virus and SPVD= Sweet potato virus disease, which is a disease complex caused by co-infection of SPFMV and SPCSV

Materials and methods

Comment 8: Line 121: Which of the treatments was carried out by the I. setosa plants and how were others treatment carried out?

Response 8: Comment has been addressed for more clarity. I.Setosa a wild relative of sweetpotato, was inoculated with the three different virus treatments (SPVD, SPFMV and SPCSV). Infected scions were then used to graft inoculate the different cultivars. This is because I. Setosa is very susceptible to sweetpotato viruses and is used to increase viral load in the plant. For this reason, I. Setosa is used in both virus diagnostics and virus inoculation by grafting.

Comment 9: Line 129 contradicts line 121. How many treatments were used in the study and what are they?

Response 9: There three virus treatments and controls for every treatment, therefore making 4 in the calculation of samples. However, the sentence has been re-written for clarity.

Results

Comment 10: Lines 254 – 262: results presented here look more general, I think readers may need to know more about the

Response 10: Authors did not quite get the comment here, so it is not addressed.

Comment 11: Lines 268-269: This is not true. By observing figure 2B, the mock control does not clearly show a distinct cluster from viral treatments.

Response 11: I tried to present all sweetpotato cultivars in the figure. And as you see from the text, ‘Tanzania’ and ‘New Kawogo’ exhibited very limited symptoms. The message to convey here is the symptom expression between cultivars. However, photos have been replaced.

Discussion

Comment 12: Line 570 and elsewhere throughout the manuscript: full name should appear just once in core text of the manuscript (weeks post-infection WPI = WPI)

Response 12: Authors agree with the reviewer and comment has been addressed

Comment 13: Line 573: reference after ‘defense activation’

Response 13: Reference has been inserted and sentence re-written

Comment 14: Line 694-696: The sentence looks grammatically incorrect; please check

Response 14: The sentence has been re-written to address reviewer’s comment.

References

Comment 15: Lines 705-708: remove capital letters within sentences and keep constancy with other references, same at lines 717-718, 818-815

Response 15: Comment has been addressed

Reviewer 3 Report

Comments and Suggestions for Authors

For my comments and suggestions - please see the attached script (pdf) file.

Author Response

Line 28: In which region, specify please

Response: Author have addressed this comment

Line 31: Why were the three varieties Beauregard, Tanzania and New kawogo

Response: To compare identify genes responsible for susceptibility or resistance to SPVD, these three cultivars were chosen because of their phenotypic response to SPVD. Beauregard is highly susceptible; Tanzania is moderately tolerant and New kawogo is highly tolerant. There is no sweetpotato cultivar that is resistant to SPVD worldwide.

Line 38: Some of the transcriptomic data results should be mentioned in the abstract

Response: Authors agree with reviewer and transcriptome data has been added to the abstract.

Line 61: mention symptoms of SPVD

Response: Authors agree with reviewer and comment has been addressed.

Line 62: What is is the typical disease incidence and associated yield loss caused by single virus infections in sweet potato?

Response: Comment has been addressed.

Line 63: How much annual losses SPVD causes?

Response: Annual losses caused by SPVD varies worldwide because the incidence of SPVD incidence varies globally.  However, yield losses are measured in % of yield loss. As indicated SPVD can cause up to 98% yield loss in very high incidences. Comment has been addressed.

Line 118: What was the source of the inoculum, and how was the SPVD infection, including the presence of both viruses, confirmed.

Response: Virus inoculum mentioned in the paper was obtained from isolates SPFMV Piu3 isolate (GenBank ID: FJ155666), SPCSV m2 _47 isolate (GenBank ID: HQ291259 for RNA1 and HQ291260 for RNA2) that have maintained in the screen house. Virus presence was confirmed by mapping RNA-Seq reads to the corresponding virus genomes. This study is continuation of transcriptome study. Virus detection results are being analysed; however, results form Beauregard were published by Bednarek et al. 2021. Results have not shown in this publication.

Line 121-125: The study refers to symptoms potentially caused by viral infection but does not present confirmatory diagnostic evidence via RT-PCR, ELISA, or sequencing. Symptomatology alone is unreliable for virus identification.

Response: As mentioned in the previous comment, virus infection was detected by NGS, however, only results from Beauregard have been analysed and published (Bednarek et al. 2021). Analysis of virus infection in Tanzania and Beauregard is being carried out and this information will be published

Line 132: Reference required

Response: Reference has been inserted

Line 142: RNA-Seq data for other two cultivars was not deposited in the databank?

Response: RNA-Seq data for Tanzania and New Kawogo have not yet been deposited in the GenBank, however, sequencing data has been stored in a server at Cornel University.

Line 236: Do all three cultivars, with varying resistance/tolerance, showed highest symptoms?

New Kawogo (highly tolerant) did not exhibit any clear symptoms (observed in only 2 plants). Tanzania (moderately tolerant) had symptoms at 3WPI but later recovered. Symptoms were more distinctly expressed in Beauregard (susceptible).

Line 238: Symptoms are not clear in the figure

Response: Thank for your comment. We currently don’t have single photos. Have reached out to co-authors in Lima and will address that comment as soon as I get more photos.

Line 240: How SPVD-associated viruses’ infection was confirmed in the plants? This is a major drawback in the study.

Response: As mentioned in previous comment, SPVD associated with virus infection was confirmed by RNA-Seq.

Line 250: Show individual leaves with characteristic symptoms compared to leaves from healthy and mock-inoculated control plants.

Response: Thank you for your comment, however we do not have photos of single leaves.

Line 258: Mention this in MM section

Response: Comment has been addressed.

Line 306: Here Cv. Tanzania performed way better - but it was excluded from molecular marker analysis, why?

Response: Tanzania is a moderately tolerant to SPVD. Molecular markers for Tanzania have been identified in this study, however; the study focused more on the susceptible cultivar Beauregard and highly tolerant cultivar New Kawogo to clearly identify distinct genes of response in different cultivar.

Line 326: Follow my previous comment about the exclusion of this cultivar from a analysis

Response: Comment has been addressed by authors in previous comment

Lines 350-351: While the collective GO term analysis provides valuable insights, I strongly recommend supplementing this with cultivar-specific GO enrichment analyses. This approach is essential to:

Identify unique defense pathways activated in each genotype (e.g., redox regulation in 'New Kawogo' vs. hormone signaling in 'Tanzania'),

Clarify genotype-dependent resistance mechanisms underlying phenotypic differences,

Validate whether key pathways identified in the aggregate analysis are consistently deployed or cultivar-specific.

Comparative visualization (e.g., heatmaps of enriched pathways by cultivar) would powerfully illustrate how resistance strategies diverge at the molecular level. 

Response: Authors appreciate your suggestions and would very much like to carry out the recommended analysis. However, the allotted timeframe of 10 days to respond to the reviewers’ comments does not allow us to conduct a thorough analysis.

Line 394: To increase clarity, authors should make this sentence more specific by including the cultivar and time point

Response: The RNA silencing pathway was activated in Beauregard infected with SPVD and SPCSV at 3 weeks.

Line 413: Relocating this to the beginning of the Results section will help readers more easily grasp the complete story early in the manuscript.

Line 241-243: This cultivar is regarded as Tolerant. Does this not contradict to its tolerance ability?

Response: This does not contract its tolerance as Tanzania is a moderately tolerant cultivar, not highly tolerant.

Line 451-453: Better rewrite this sentence in more concise way

Response: Authors agree and comment has been addressed

Line 478: it has already been abbreviated

Response: Comment has been addressed by authors

Line 554-551: Redundant to last paragraph

Response: Paragraph has been deleted to avoid redundancy

Line 561-562: Abbreviated already, no need to repeat it multiple times

Response: Authors do agree and comment has been addressed

Line 569: This is a substantial claim, yet the symptoms depicted in the figure are ambiguous

Response: New kawogo (highly tolerant) did not exhibit symptoms across all time points except in only two plants and Tanzania exhibited virus symptoms at 3 and 6 WPI but later recovered by 12 WPI. Beauregard (susceptible) exhibited the highest symptoms as seen in line 236-248. Authors will address this comment as soon as we get more photos from one of our co-authors

Lie4 614- 624: This should be interpreted and discussed with reference to prior studies conducted under comparable condition

Response: Comment has been addressed with references inserted

Lline 636-637: "Tanzania" deployed more genetic resources (highest number of DEGs) than resistant "New Kawaogo" but still a moderately tolerant? it should be discussed here

Response: Phenotypically, Tanzania id moderately tolerant to SPVD. However, as seen from the symptoms, it presented symptoms at 3- and 6- WPI and eventually recovered. Expression of highest number of DEGs, indicates a high transcriptomic response to treatment.

Line 646-647: Did the authors attempt to assess and compare yield or other quantifiable attributes of these cultivars during the course of the study?

Response: Authors did not attempt to address any quantifiable attribute since the experiment was carried out in the screen house. Sweetpotato conserved in screen houses do not usually root.

Round 2

Reviewer 1 Report

Comments and Suggestions for Authors

I appreciate the authors for doing the revision and sending response to the queries. However, I do not have a high degree of satisfaction towards the responses that authors have provided.

  1. Authors have presumed that in their experiments the virus(es) is present without doing any test for the virus(es) at molecular level (PCR or qPCR). How would you make sure that each stage your samples were having virus infection. This is clearly unacceptable for scientific publication. Mere building up on previous work does not mean that you don't have to prove or present the evidence in your current paper. I would highly appreciate if you could present this (at least detection, Viral load you can publish in other paper) data.
  2. Authors response to comment 2 is also not satisfying. I guess the authors failed to understand the query (or I failed to clearly state the query). My question is why scions were used (scions are upper part in grafting). If you are using Ipomoea setosa as infected scion then that means that you are taking samples (for trasncriptomic analyses) from I. setosa not the other varieties that you have mentioned. In case the authors think I am mis-reading the text please make it clear in your manuscript. My thinking is that for virus inoculation to test samples you would use infected rootstock. However, in case you want to test that your rootstock (any variety) is already virus infected than you can use indicator host (I. setosa) as scion. 
  3. In the revised version I don't see the changed images, they are still the same as in the first version. It would be better if you could show healthy and infected plants together in one panel. It's a humble request tp authors that do not presume that readers will automatically find the differences in symptoms, you should put extra effort in showing the differencing (or at least labelling the samples). This is clearly not a good representation of results. 

Author Response

Comment 1: Authors have presumed that in their experiments the virus(es) is present without doing any test for the virus(es) at molecular level (PCR or qPCR). How would you make sure that each stage your samples were having virus infection. This is clearly unacceptable for scientific publication. Mere building up on previous work does not mean that you don't have to prove or present the evidence in your current paper. I would highly appreciate if you could present this (at least detection, Viral load you can publish in other paper) data.

Response 1: Thank you for your detailed feedback and for clarifying your concerns. Comments have been addressed as requested by reviewer.

Comment 2: Authors response to comment 2 is also not satisfying. I guess the authors failed to understand the query (or I failed to clearly state the query). My question is why scions were used (scions are upper part in grafting). If you are using Ipomoea setosa as infected scion then that means that you are taking samples (for trasncriptomic analyses) from I. setosa not the other varieties that you have mentioned. In case the authors think I am mis-reading the text please make it clear in your manuscript. My thinking is that for virus inoculation to test samples you would use infected rootstock. However, in case you want to test that your rootstock (any variety) is already virus infected than you can use indicator host (I. setosa) as scion. 

Response 2: Thank you for your detailed feedback and for clarifying your concerns regarding the grafting procedure. We apologize for any misunderstanding in our previous responses to Comments, and we appreciate the opportunity to address this more precisely. To ensure clarity, we have revised the relevant sections of the manuscript (in the Methods section) to explicitly describe the grafting protocol, the roles of the scion and rootstock, and the source of samples for transcriptomic analyses.

In brief, the goal of our experiment was to inoculate virus-free sweetpotato cultivars with specific virus treatments (SPVD, SPFMV, and SPCSV) to study their transcriptomic responses to infection. I. setosa, a wild relative highly susceptible to sweetpotato viruses, was first graft inoculated with the viruses to serve as an infected source plant with high viral load. Scions from these infected I. setosa plants were then grafted onto healthy plants of the target sweetpotato cultivars (e.g., Beauregard, Tanzania, etc.). This top-grafting approach allows systemic transmission of the viruses from the infected scion downward into the rootstock, as is standard for virus inoculation in sweetpotato research (e.g., as described in Moyer and Salazar, 1989; and recent studies such as Bednark et al., 2021 and Kreuze et al., 2020).

We did not use infected rootstocks for inoculation, as that would not align with our objective of introducing controlled virus infections into initially virus-free cultivars. After successful graft-inoculation and confirmation of infection via symptom observation and sRNA analysis, leaf samples for RNA extraction and transcriptomic sequencing were collected exclusively from the infected sweetpotato (i.e., the target cultivars), not from the I. setosa scions. This ensures that our analyses reflect the molecular responses of the sweetpotato varieties themselves.

Comment 3: In the revised version I don't see the changed images, they are still the same as in the first version. It would be better if you could show healthy and infected plants together in one panel. It's a humble request tp authors that do not presume that readers will automatically find the differences in symptoms, you should put extra effort in showing the differencing (or at least labelling the samples). This is clearly not a good representation of results. 

Response 3: Images have been changed to address reviewer’s comment 

Reviewer 3 Report

Comments and Suggestions for Authors

I appreciate the authors' detailed responses. To finalize the manuscript, please integrate the changes from the 'Response to Reviewers' document into the main text to improve clarity for the reader. For example related to;
1. Data submission
2. Comfirmation of source of inoculum
3. Mention previous study (Bednarek et al. 2021) at appropriate place

Additionally, please add the missing plant figures. These revisions are required for acceptance

Author Response

Comments and Suggestions for Authors

I appreciate the authors' detailed responses. To finalize the manuscript, please integrate the changes from the 'Response to Reviewers' document into the main text to improve clarity for the reader. For example related to;

  1. Data submission
    2. Comfirmation of source of inoculum
    3. Mention previous study (Bednarek et al. 2021) at appropriate place

    Additionally, please add the missing plant figures. These revisions are required for acceptance

Response: Thank you for your thorough review and constructive feedback on our manuscript. We greatly appreciate your time and suggestions, which have helped strengthen our work. We have carefully addressed each of your comments and incorporated the requested changes into the revised manuscript. Below, we outline how we have responded to your specific points:

  1. Data Submission: We have integrated the details from the 'Response to Reviewers' document into the main text to clarify the data submission process. In the (Data Availability or Methods section), we have added the following statement: "Raw RNA-seq reads for ‘Beauregard’ were deposited in the NCBI Sequence Read Archive (SRA) under the accession number PRJNA649319. Raw RNA-seq reads for ‘New Kawogo’ and ‘Tanzania’ were uploaded onto a server at Cornell University under the RNA-seq data processing for the Genomic Tools for Sweetpotato Improvement (GT4SP) project, funded by the Bill & Melinda Gates Foundation (Contract ID: OPP1052983), for analysis." This addition ensures that readers have clear and comprehensive information about the accessibility and location of all raw data.
  2. Confirmation of Source of Inoculum: We have added analysis of sRNA analysis to confirm infection in sweetpotato plants after grafting with virus infected I. Setosa as the source of the inoculum, as outlined in our previous response. This addition enhances the reproducibility and clarity of our methodology.
  3. Reference to Bednarek et al. (2021): We have included a reference to Bednarek et al. (2021) at the appropriate context, as suggested. This citation strengthens the connection to prior work and provides a more comprehensive background for our study.
  4. Missing Plant Figures: The missing plant figures have been added to the manuscript. These figures are now properly labeled and referenced in the text to improve the visual representation of our findings.

We believe these revisions address all concerns and enhance the clarity and quality of the manuscript. The updated manuscript, with all changes tracked for your convenience, is attached for your review. Please let us know if any further adjustments are needed.

Once again, thank you for your valuable feedback and for considering our work for publication.